# Multipurpose silicon photonics signal processor core

Daniel Pérez[1], Ivana Gasulla[1], Lee Crudgington[2], David J. Thomson[2], Ali Z. Khokhar[2], Ke Li[2], Wei Cao [2], Goran Z. Mashanovich[2,3] & José Capmany[1]

Integrated photonics changes the scaling laws of information and communication systems offering architectural choices that combine photonics with electronics to optimize performance, power, footprint, and cost. Application-specific photonic integrated circuits, where particular circuits/chips are designed to optimally perform particular functionalities, require a considerable number of design and fabrication iterations leading to long development times. A different approach inspired by electronic Field Programmable Gate Arrays is the programmable photonic processor, where a common hardware implemented by a two-dimensional photonic waveguide mesh realizes different functionalities through programming. Here, we report the demonstration of such reconfigurable waveguide mesh in silicon. We demonstrate over 20 different functionalities with a simple seven hexagonal cell structure, which can be applied to different fields including communications, chemical and biomedical sensing, signal processing, multiprocessor networks, and quantum information systems. Our work is an important step toward this paradigm.

[1] ITEAM Research Institute, Universitat Politècnica de València, Camino de Vera s/n, Valencia 46022, Spain. [2] Optoelectronics Research Centre, University of Southampton, Highfield, Southampton SO17 1BJ, UK. [3] School of Electrical Engineering, University of Belgrade, 11120 Belgrade, Serbia. Correspondence and requests for materials should be addressed to J.C. (email: jcapmany@iteam.upv.es)

Photonic integrated circuits (PICs)[1–10] combine the unique properties of optical waveguides such as low propagation losses, absence of diffraction, high power confinement, low crosstalk, and immunity to electromagnetic interference with other highly desirable features such as small footprint, compactness, stability, reduced power consumption, and the possibility of low-cost fabrication.

These features make photonic integration ideal for a wide variety of emerging applications with potential massive impact. In high-speed fiber communications, PICs enable flexible and reconfigurable transceivers and multiplexers supporting all the multiplexing domains (wavelength, polarization, and space)[11, 12]. PICs will play a critical role in future 5 G communication systems[13–15], the Internet of Things[16] and advanced civil radar systems[17], where broadband and upgradable interfaces between the fiber and wireless network segments will be required at the base stations. In chemical and biomedical sensing, PICs enable the concept of photonic lab on a chip[18, 19]. PICs also enable high-speed[20] signal processing operations[21–24], reconfigurable interconnections between processing and memory units in advanced multiprocessor computing systems and data centres[25–27] as well as quantum logic gates[28–30].

The so-called application-specific photonic integrated circuit (ASPIC) paradigm has been dominant so far in integrated optics[31]. In this approach, a particular circuit configuration is designed to optimally perform a particular functionality in terms of propagation losses, power consumption, footprint and number of components. An ASPIC design requires, however, a considerable number of design and fabrication iterations leading to a long time for development[32]. Furthermore, despite the fact that a reconfigurable processor implementing signal integration, differentiation and Hilbert transformation using an InP ASPIC has been recently reported[33], multifunctional ASPICs are difficult to design keeping the same layout.

An alternative to ASPICs is to consider a general-purpose processor architecture that can be integrated on a photonic chip, featuring single and/or multiple input/output operation and being capable of performing different signal processing tasks by programming of its electronic control signals[34, 35]. Several authors[36–38] have reported seminal theoretical work proposing different architectures and design principles based on the cascade of beamsplitters or Mach Zehnder Interferometers (MZIs) that incorporate phase tuning elements, which enable independent control of amplitude and phase of light. These configurations are targeted in particular for multiple input/multiple output feedforward linear optics transformations.

A more versatile architecture can be obtained by following similar principles as those of the Field Programmable Gate Arrays in electronics[39]. The core concept is to break down complex circuits in a large network of identical two-dimensional (2D) unit cells implemented by means of a MZI waveguide mesh or lattice. Zhuang and co-workers[39] have pioneered the field by proposing a programmable optical chip architecture connecting MZI devices in a square-shaped mesh network grid. The distinctive feature of this approach is that it enables both feedforward and feedbackward configurations, selecting the adequate path through the mesh and providing independent tuning of circuit parameters to complex valued coefficients by introducing phase tuning elements in both arms of the MZIs to enable independent control of amplitude and phase of light at coupler outputs[39, 40]. The structure, fabricated in $Si_3N_4$ consisted in two square cells and was employed to demonstrate simple finite (FIR) and infinite (IIR) impulse response filters with single and/or double input/output ports.

Inspired by this approach, other mesh topologies were proposed and compared to the square grid[41]. In particular, hexagonal and triangular-shaped meshes[41] feature improved performance in terms of spatial tuning reconfiguration step, reconfiguration performance, switching elements per unit area and losses per spatial resolution[41] (see Supplementary Note 1). In addition, the hexagonal mesh enables the simplest implementation of both multiport interferometers and classic FIR and IIR photonic circuits.

Here, we report the design, fabrication and experimental demonstration of a silicon photonics multipurpose processor core based on an integrated hexagonal waveguide mesh. The mesh is composed of 7-hexagonal MZI waveguide cells. We demonstrate the implementation of over 20 different configurations of photonic circuits ranging from simple single-input/single-output FIR filters, optical ring resonators (ORRs), coupled resonator waveguides (CROWs), side-coupled integrated spaced sequences of optical resonators (SCISSORs) and ring loaded MZIs to multiple-input/multiple-output linear optic $2 \times 2$, $3 \times 3$, and $4 \times 4$ transformations including Pauli Matrices and a C-NOT gate. The experimental demonstration of this multifunctional integrated waveguide mesh photonic processor core includes both classical FIR and IIR signal processing functions as well as multiport linear optics operations.

## Results

**Processor and waveguide mesh layout**. The architecture concept of the software-defined photonic processor is shown in Fig. 1a. Its central element is the optical core, where the main signal processing tasks are carried in the photonic domain. We have proposed its implementation by means of a photonic 2D hexagonal waveguide lattice[41]. Figure 1b illustrates this mesh topology. Each hexagon side or basic unit length (BUL) is composed of two close waveguides, and the connection between them is controlled by means of a tuneable basic unit (TBU) that is implemented by 3-dB MZIs (Fig. 1c). Through external electronic control signals applied to heaters deposited on top of each MZI arm, each TBU can be configured to operate as a directional coupler or simply as an optical switch in a cross or bar state providing amplitude- and phase-controlled optical routing (see Supplementary Note 2). Using this principle, this common architecture can be reconfigured to support any kind of linear transformation between multiple input and multiple output waveguide ports, as in self-configuring universal linear components, much in the same way as a FPGA operates in electronics. Figure 1c shows the layout of the 7-cell lattice reported in this paper together with the details and relevant parameters of its TBUs.

**Filter synthesis algorithms and scalability**. The proposed hexagonal waveguide mesh is per se a programmable hardware platform supporting multiple configurations and not a specific layout. This means that it can be programmed to emulate the specific hardware configurations of a wide variety of signal processor architectures, including, among others, traditional feedforward/feedbackward FIR and IIR impulse response filters as well as multiple-input/multiple-output optical linear transformers. It is well known from signal processing theory[42] that given a particular hardware configuration the range of transfer functions that can be synthesized with it is limited. In other words, it is not possible to design a general and well-defined synthesis algorithm accounting for all the possible filter realizations. Therefore, not such synthesis algorithm can be developed for the hexagonal waveguide mesh either. However, the available synthesis methods for the specific hardware configurations that can be emulated using the waveguide mesh can be applied by developing a suitable procedure, which translates the results

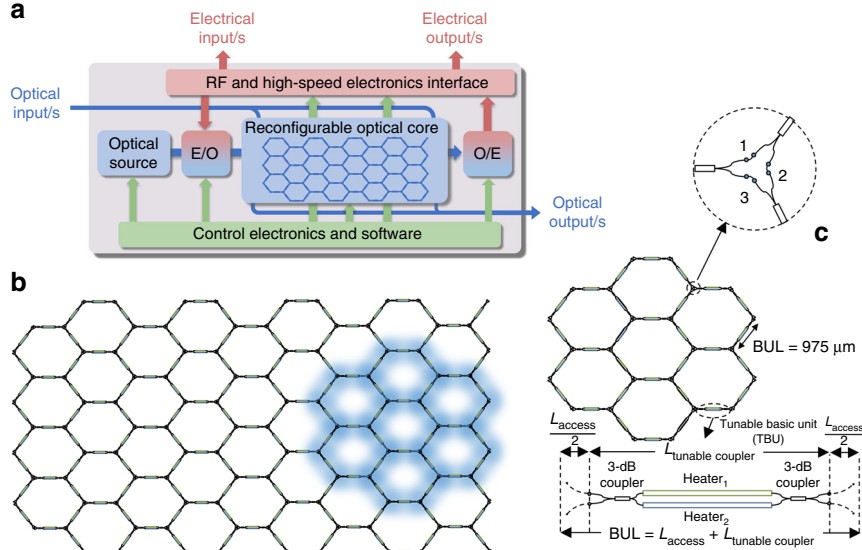

**Fig. 1** Software-defined general-purpose photonic processor and its waveguide mesh reconfigurable core. **a** Architecture of the processor showing the reconfigurable core as its central element and the different possible electrical, optical and control input/output signals (E/O: External modulator. O/E: Optical receiver). **b** Schematic of the hexagonal waveguide mesh; a 7 unit cell structure is marked in *blue*. **c** Layout detail of the 7-cell hexagonal waveguide mesh designed and fabricated, including a zoom (*lower*) of a hexagon side of length 1 basic unit length (*BUL*) implemented by means of a 3-dB Mach Zehnder Interferometer (*MZI*) and a zoom (*upper*) of an optical interconnection node. TBU, tuneable basic unit

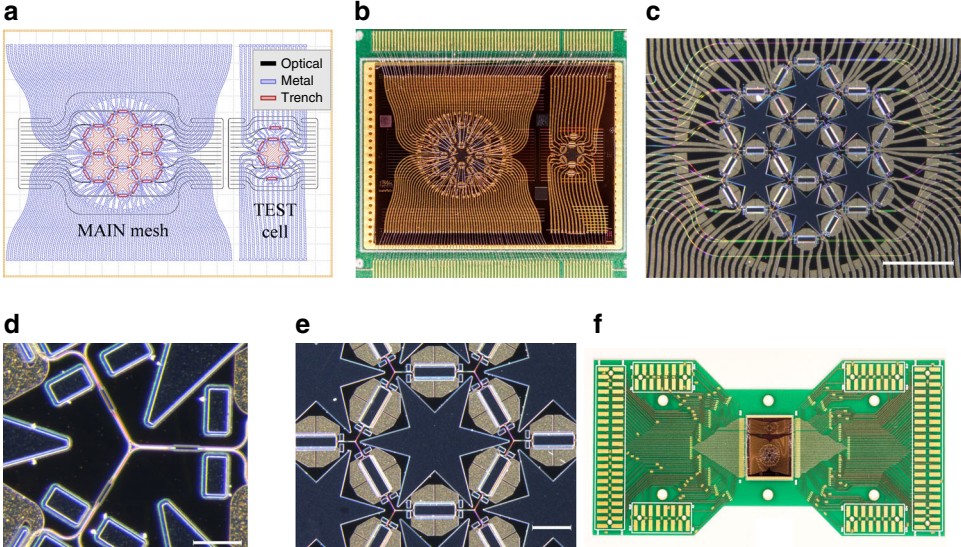

**Fig. 2** Fabricated hexagonal waveguide mesh chip. **a** Design layers (optical, electrical, and thermal) of the 7-cell hexagonal waveguide mesh and the auxiliary test cell. **b** Fabricated silicon on insulator (*SOI*) chip of footprint 15 × 20 mm. **c** Zoomed vision of the 7-cell hexagonal waveguide mesh. *Scale bar* of 2 mm. In the *right bottom* corner **d** zoomed image of an optical interconnection node of three tuneable basic units (TBUs). *Scale bar* of 100 μm. In the *right bottom* corner **e** zoomed image of a single hexagonal cell showing the Mach Zehnder Interferometer (*MZI*). *Scale bar* of 500 μm. In the *right bottom* corner, tuning heaters, and star-type thermal isolation trenches. **f** Printed circuit board with the waveguide mesh chip mounted and wired bonded

provided by the synthesis equations into specific parameter values of the MZI that are needed to implement the waveguide coupling points in the emulated layout. We have found that this is possible for all main discrete-time signal processing hardware configurations employed in practice as discussed below, all of which are scalable.

For example, FIR filters are based either on cascades/lattices of 3-dB tuneable MZIs or in transversal filter configurations. For both FIR filter alternatives, synthesis, and recursive scaling algorithms have been developed in the literature[42, 43] that are directly applicable since the hexagonal waveguide mesh can directly implement both 3 dB-tuneable MZI cascade lattices and

transversal filter configurations. For IIR filters, either simple/compound optical ring cavities of ring-loaded 3-dB tuneable MZI cascades are employed. Again, synthesis algorithms have been reported in the literature[42, 44] that are directly applicable since the hexagonal waveguide mesh can directly implement either simple or multiple cavity ring filters or ring-loaded 3-dB tuneable MZI cascades.

In the case of multiple-input/multiple-output optical linear transformers, detailed synthesis and recursive scaling algorithms have been reported by Miller for triangular configurations[36] and by Clements et al.[38] for rectangular configurations, which, as mentioned above, need to be adapted to the hexagonal waveguide

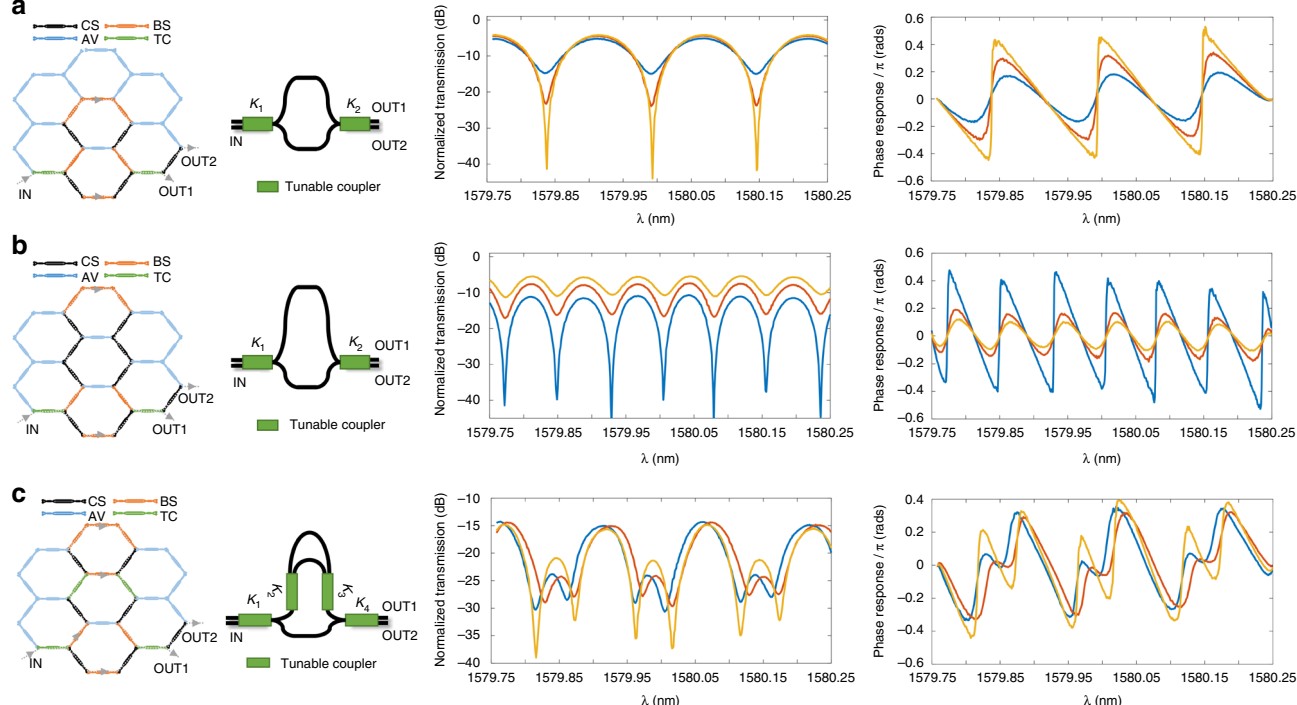

**Fig. 3** Experimental results for tuneable unbalanced Mach Zehnder Interferometers and finite impulse response filters. Waveguide mesh connection diagram, circuit layout and measured modulus, and phase transfer function for different values of the coupling constants $K_1$ and $K_2$ in the case of **a** a 4-BUL unbalanced Mach-Zehnder Interferometer (UMZI) filter; **b** an 8-BUL UMZI filter; **c** a 4-BUL basic delay 3-tap transversal filter. For each case, the first column shows the 7-cell hexagonal waveguide mesh configuration, where each Mach Zehnder Interferometer (MZI) device is represented by a given color depending on whether it is activated as a cross (black) or bar (orange) switch, a tuneable coupler (green) or not used/available (blue). The second column shows the layout of the implemented structure, while the third and fourth columns show, respectively, the measured modulus and corresponding phase (calibrated by the shortest path) for the synthesized configuration where the input is in the IN port and the output is the OUT1 port. Measured curves are displayed for different values of the coupling constants $K_1$ and $K_2$, which are tuned by changing the injection currents to the heater elements of the input and output MZI devices of the UMZI. Changing these values alters the position of the zero in the UMZI transfer function bringing it closer or farther to the unit circle[42]. The closer the zero is to the unit circle, the deeper are the notches in the transfer function are and the higher is the phase shift step in the transfer function is ref. [42]. BUL, basic unit length; CS, cross state; BS, bar state; AV, available; TC, tuneable coupler

mesh configuration. We provide these adaptations in the Supplementary Note 3.

**Chip fabrication and testing**. The hexagonal waveguide mesh reconfigurable core is fabricated in silicon on insulator (SOI) and wire-bonded to a chip carrier for experimental demonstration (see Methods). The design process for which the layout is shown in Fig. 2a involved three different steps for the optical waveguide layout, the metal electrodes required to tune the MZIs and the trenches for thermal isolation. An auxiliary test structure was included for characterization. Figure 2b shows the fabricated chip, which occupies a surface of $15 \times 20\ mm^2$ and includes 30 MZIs, 60 thermal tuners, 120 pads and features 24 optical input/output ports. The test structure has an additional 8 MZIs (6 + 2), 16 thermal tuners and 32 TBUs. Figure 2c–e display zoomed views of the 7-hexagonal mesh an optical interconnection node of three tuneable basic units and a unit cell respectively, where the MZI and the tuning electrodes are clearly distinguishable. The hexagon BUL is 975 μm, corresponding to a delay of 13.5 ps. Figure 2f shows the chip mounted on a printed circuit board (PCB), which occupies a surface of $60 \times 120\ mm^2$.

Full static testing of the device including propagation, bend and insertion losses as well as thermal stability was carried using the test cell (see Methods). We measured the spectral region of operation for the input/output grating coupler devices and found that the optimum performance was in the (1580 ± 15 nm) range

rather than the targeted 1550 nm. We obtained full calibration curves for the coupling constants and phase shift versus injected current for all the 30 MZIs in the structure, as well as a detailed characterization of the thermal crosstalk (see Supplementary Note 4). Despite its simple configuration, the 7-hexagonal cell lattice can implement over 100 different signal processing configurations by suitably tuning its MZI elements (see Supplementary Note 5). We now describe a selection of some relevant results in terms of increasing complexity.

**Basic tuneable MZI and FIR filters**. Unbalanced Mach-Zehnder interferometers (UMZIs) are 2-input/2-output periodic notch filters that constitute the basic building blocks for lattice and FIR transversal filters[42]. UMZIs find multiple applications[43], including linear phase filters, multi-channel selector biosensors and group delay compensators to name but a few. By suitably tuning the MZI devices in the 7-cell waveguide mesh, we have been able to implement UMZI devices with path unbalances given by 2, 4, 6, and 8 BULs, limited by the number of current sources presently available for the moment of measurement. Figures 3a, b show, as an example, the results for the 4- and 8-BUL UMZI cases. Note that the periodicity in the transfer function (free spectral range or FSR) changes according to the path unbalance (18.4 GHz for the 4-BUL UMZI and 9.2 GHz for the 8-BUL UMZI). In each case, we compare the experimental results with those provided by the theoretical expressions of the transfer functions obtaining an excellent agreement. We also checked the tuning of the notch

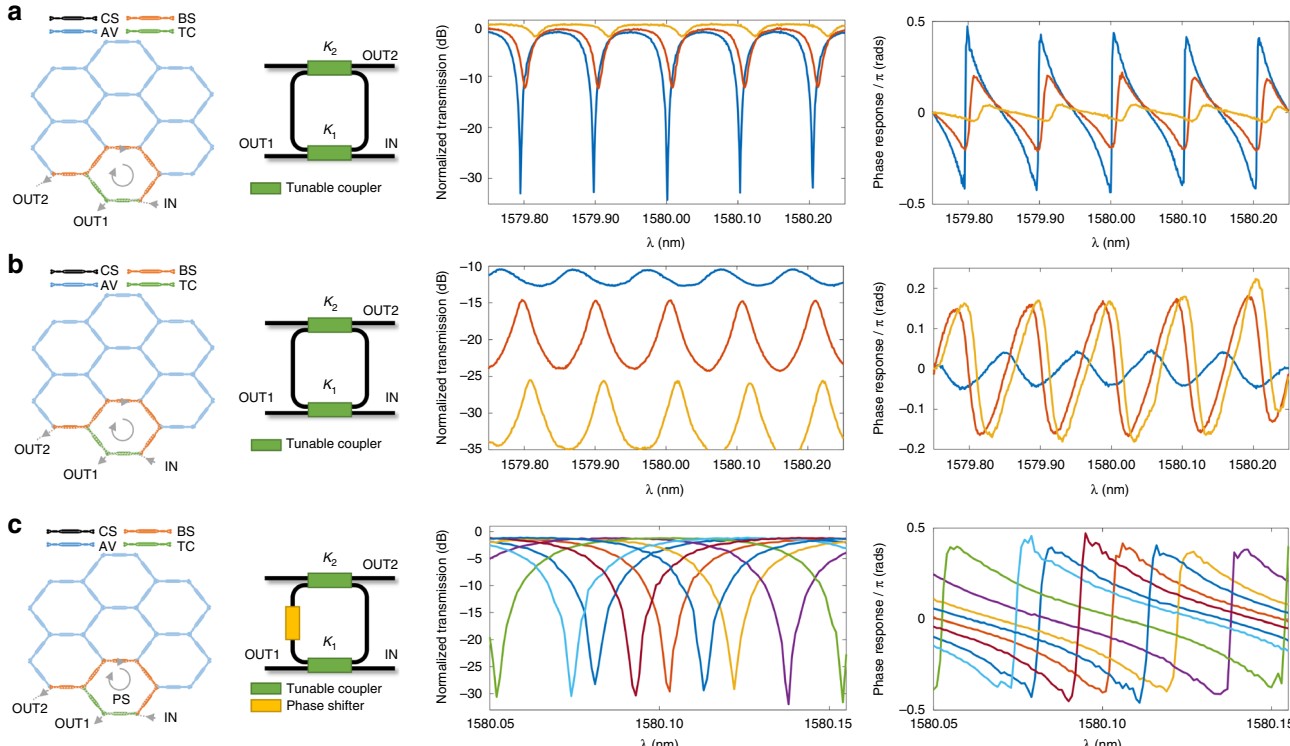

**Fig. 4** Experimental results for 6-BUL ring resonator infinite impulse response and combined finite impulse response and infinite impulse response filters. Waveguide mesh connection diagram, circuit layout and measured modulus, and phase transfer function for **a** a 6-BUL optical ring resonator (*ORR*) infinite impulse response (*IIR*) filter for different values of the coupling constants $K_1$ and $K_2$; **b** a 6-BUL ORR finite impulse response (*FIR*) + IIR filter for different values of the coupling constants $K_1$ and $K_2$; **c** a 6-BUL ORR IIR filter along a full spectral period for different values of the optical ring resonator round-trip phase shift. BUL, basic unit length; CS, cross state; BS, bar state; AV, available; TC, tuneable coupler

position over a complete spectral period by proper phase shifting in one of the UMZI arms[42] (see Supplementary Note 6 for more detailed information).

Since the waveguide mesh has a limited number of cells, we were unable to implement a lattice filter by serially cascading UMZI units. However, we could implement a 3-tap transversal filter by the parallel cascade of UMZI units. The results are shown in Fig. 3c. The transversal filter is band-pass periodic, as expected, with a FSR of 18.4 GHz, given by the inverse of the basic delay, which in this case was 4 BUL. Changing the values of $K_1$ to $K_4$ we tuned the positions of the two zeros provided by the structure and, therefore, reconfigure its transfer function.

**Basic tuneable ring cavities and IIR filters**. Ring cavities are either 1-input/1-output or 2-input/2-output periodic filters. In the first case, they implement all-pole infinite impulse response (IIR) notch filters, while in the second they can implement both IIR notch and FIR + IIR bandpass filters[42]. They constitute the basic building blocks for more complex filter designs such as CROWs and SCISSORs. Ring cavities find multiple applications[44] including integrators, differentiators and Hilbert transformers[33], dispersion compensators[45], as well as tuneable radiofrequency phase shifters, and true time delay lines[46]. By suitably tuning the MZI devices in the 7-cell waveguide mesh, we have been able to implement single ORRs with cavity lengths given by 6, 12, and 18 BULs. Figure 4 shows the relevant measured results for the 6-BUL ORR length case, (for results on the 12- and 18-BUL ORR lengths, refer to the Supplementary Note 6). Figure 4a, b show, respectively, the waveguide mesh configurations (with the MZI device status according to the color code previously described), the circuit layouts and the modulus as well as the phase shift responses of the IIR (Fig. 4a) and FIR + IIR (Fig. 4b) cases.

The measured results correspond to different values of $K_1$ and $K_2$, which settle the positions of the zero and the pole[42]. The IIR filter tunability, which is shown in Fig. 4c, is achieved by exploiting the fact that the coupling constant and the phase shift in any MZI device of the mesh can be adjusted independently. Hence, any MZI can be operated as a constant-amplitude phase shifter. Changing the injected current to the phase shifter, we can incorporate any phase shift from 0 to $2\pi$ into the ORR round-trip and tune the resonance position along a full spectral period. We compared our experimental results with those predicted by the theory showing an excellent agreement (see Supplementary Note 6).

**Complex tuneable and reconfigurable filters**. We can build more complex (multicavity) signal processing structures like CROWs[47], SCISSORs[48], and ring-loaded MZIs[49] using the former basic building blocks in the 7-cell waveguide mesh and activating more MZI devices to provide additional propagation paths. These are usually 2-input/2-output filters that are characterized by transfer functions with a higher number of zeros and poles. By suitably tuning the coupling constants, one can obtain, for instance, filters with special characteristics in the modulus[42] (flat passband) of the phase shift[44] (parabolic). The number of current sources allowed us to program 2- and 3-ORR CROWs (6-BUL ORR length), 2-ORR SCISSORs (6-BUL cavity length) and a double ring-loaded MZI.

Figure 5 shows three examples of complex filters that employ two 6-BUL ring cavities (for the rest of implemented structures, see the Supplementary Note 6). Figure 5a corresponds to a bandpass CROW structure, Fig. 5b to a notch SCISSOR filter and Fig. 5c to a double ring-loaded MZI. As in the previous cases, the first column shows the 7-cell hexagonal waveguide mesh

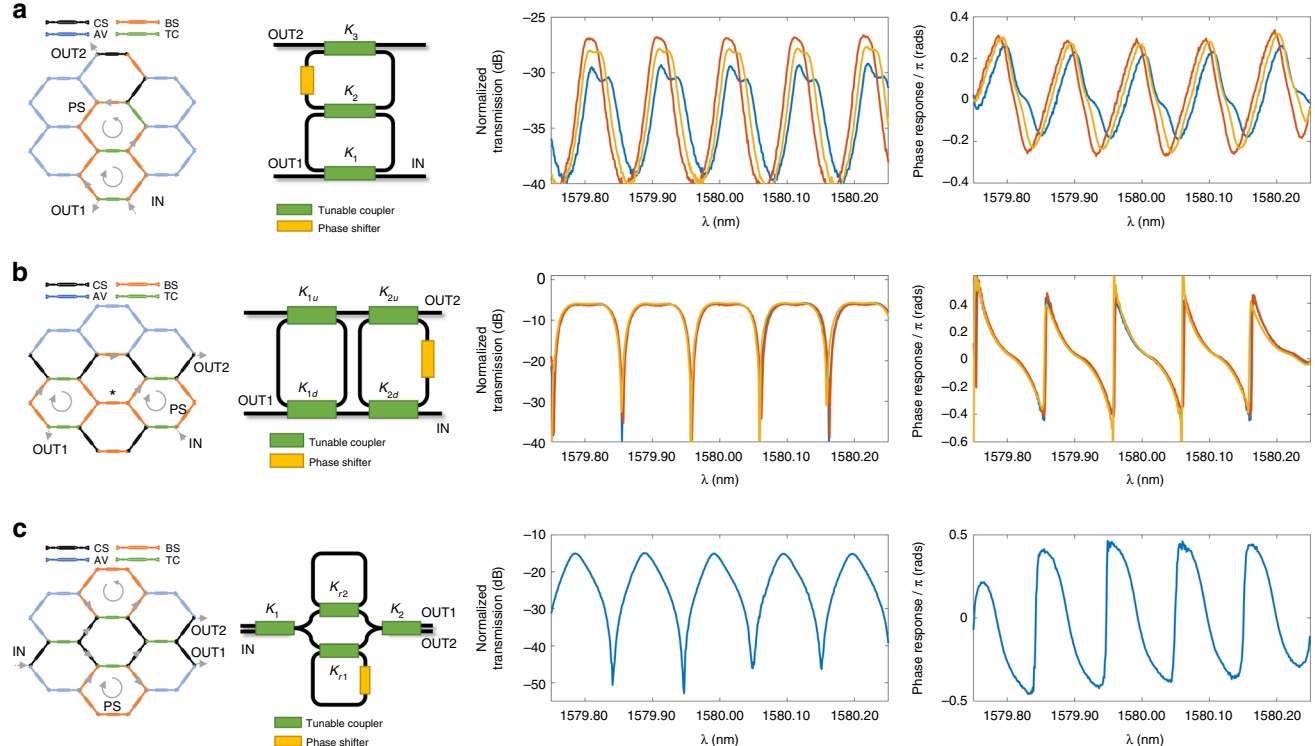

**Fig. 5** Experimental results for complex double ring-loaded 6-BUL optical ring resonator filters. Waveguide mesh connection diagram, circuit layout and measured modulus, and phase transfer function for **a** a 6-BUL double optical ring resonator (ORR) coupled resonator waveguide (CROW) filter and different values of the coupling constants $K_1$ and $K_2$; **b** a 6-BUL double ORR side-coupled integrated spaced sequences of optical resonators (SCISSOR) filter and different values of the coupling constants $K_1$ and $K_2$; **c** a 6-BUL double ORR ring-loaded Mach Zehnder Interferometer (MZI). BUL, basic unit length; CS, cross state; BS, bar state; AV, available; TC, tuneable coupler

configuration where each MZI device is represented by a given color depending on its activation state, the second column shows the layout of the implemented structure and the third and fourth columns show, respectively, the measured modulus and corresponding phase. In the measured results of Fig. 5a (input: IN, output: OUT 2), the different traces correspond to different values of the phase shifter, which move one ORR resonance with respect to the other. When the phase shift is 0, then the resonances of the two cavities are located in the same frequency and the narrowest bandpass is achieved (red trace). As a small phase shift is added to one of the cavities, one of the resonances is slightly displaced but there is still a considerable overlapping. This technique is employed to broaden the response of bandpass filters providing a controlled ripple value[42]. A similar concept is employed in the SCISSOR structure (input: IN, output: OUT 1). The phase shifter flattens the spectral region in between two consecutive notches and provides two slightly parabolic phase shifts of opposed concavity in that region, which correspond to two linear group delay regions of opposed slopes. Within this region, the structure can be employed as a tuneable dispersion compensator or as a true time delay line[44]. The TBU marked with an asterisk is an example of how TBUs can be configured in order to extract non-ideal leaking due to optical crosstalk from the circuit. Anyway, no deteriorated performance was observed for a measured optical crosstalk around 40 dB. Figure 5c shows the measured results for a double ring-loaded MZI. This structure is employed as a building block for the implementation of special configurations such as maximally flat high-order Butterworth and Chebyshev filters[42, 49].

**Multiple input/output linear optic transformation devices**. A wide variety of signal processing operations involve mode transformations, which can be described in terms of multiple

input/multiple output linear optics transformations given by an $N \times N$ unitary matrix $U$[50–52]. These include, among others, switching and broadcasting, mode combiners and splitters, and quantum logic gates. We programmed the 7-cell waveguide mesh to demonstrate several $3 \times 3$ and $4 \times 4$ linear transformations. These are relevant examples of signal processing tasks that are needed in different applications and the results are shown in Fig. 6. Figure 6a shows an example of a $3 \times 3$ column swapper between inputs 1 and 3 leaving column 2 invariant. The experimental results show a transmission ratio between desired and undesired connections in excess of 25 dB. In addition, any phase relationship between the three output modes can be selected by proper biasing MZIs M21 and M31. Incidentally, this degree of freedom in the phase can be employed to implement linear transformations for either X or Y Pauli gates between inputs 1 and 3[53] (for information on how to implement the Pauli Z gate see Supplementary Note 5), Fig. 6b shows another $3 \times 3$ transformation example. In this case, a backward input swapper where input 1 is routed to output 3, input 2 to output 1 and input 3 to output 2. Again, the experimental results show a transmission ratio between desired and undesired connections over 25 dB. The former examples are special cases that illustrate the application of the waveguide mesh as a programmable signal router.

As a final example, Fig. 6c illustrates the implementation example of a linear transformation of a C-NOT gate. This is a universal gate in quantum information and its operation can be described by a unitary $4 \times 4$ matrix[53]. Again, the results show a transmission ratio between desired and undesired connections of over 25 dB. We can program the mesh to implement this gate in a very compact layout. More complex $4 \times 4$ transformations with independent phase relationships can also be programed in the 7-cell waveguide mesh.

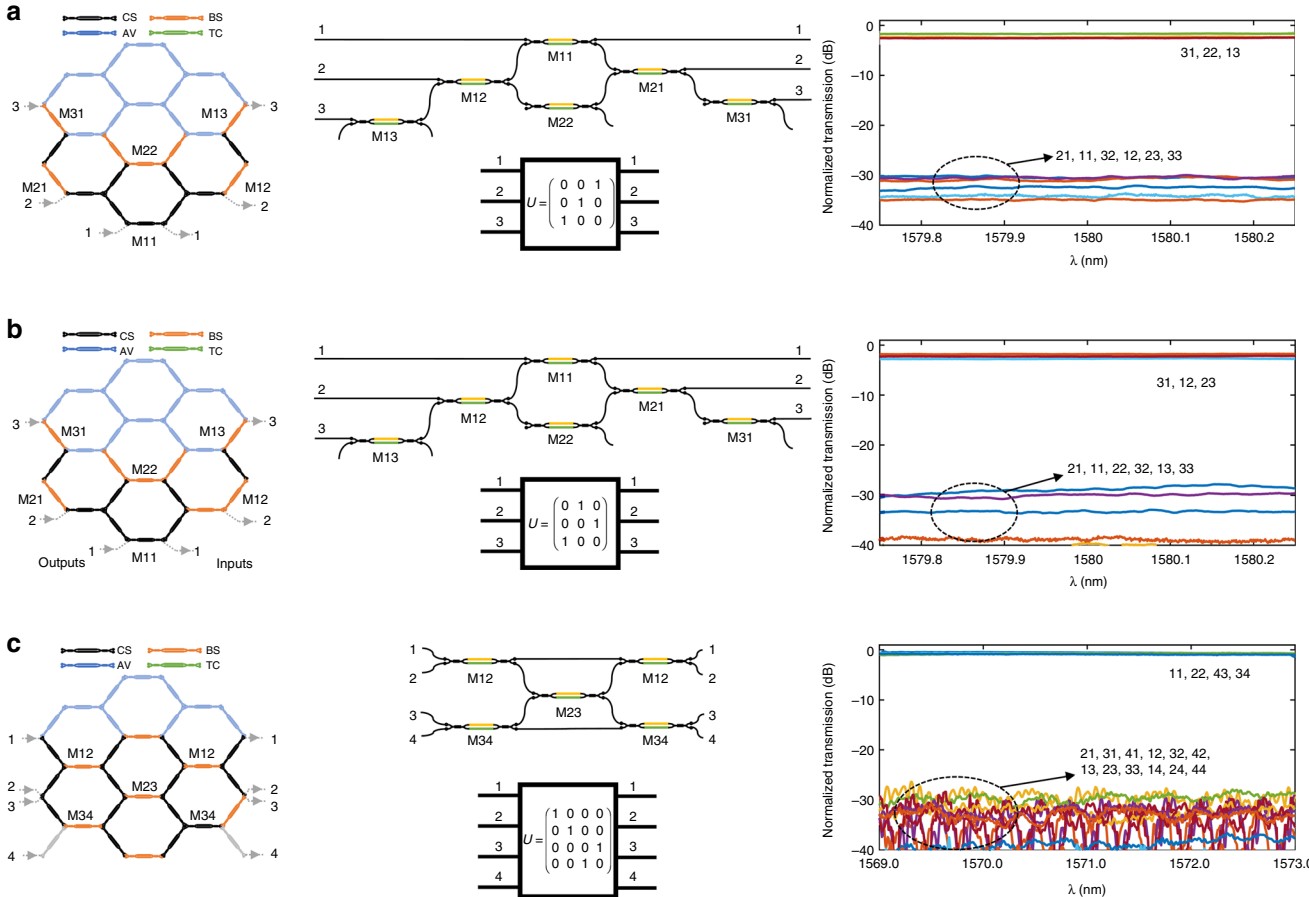

**Fig. 6** Experimental results for multiple input multiple output linear optic transformation devices. Waveguide mesh connection diagram, circuit layout, and measured modulus transfer function for **a** a 3 × 3 column swapper between inputs 1 and 3; **b** 3 × 3 backward input swapper; **c** C-NOT gate. For each case, the *first column* shows the 7-cell hexagonal waveguide mesh configuration, where each Mach Zehnder Interferometer (MZI) device is represented by a given color depending on whether it is activated as a cross (*black*) or bar (*orange*) switch, a tuneable coupler (*green*) or not used/available (*blue*). The *second column* shows the layout of the implemented structure, while the *third column* shows the spectral measurement (modulus) of all input/output port connections labeled by *XY*, where *X* and *Y* represent the output and input port numbers, respectively. CS, cross state; BS, bar state, AV, available, TC, tuneable coupler

## Discussion

The proposed waveguide mesh photonic processor can implement a wide variety of signal processing functionalities. Owing to the hexagonal-shape, even a relatively low cell count design reported here can be programed to implement over 100 different configurations (see Supplementary Notes 4 and 5). In practice, we were limited by the number of available current sources that are required to tune the TBUs, which restricted the number of configurations that we could demonstrate to 21. More complex structures (for instance, cascade UMZI lattice filters and more complex quantum logic gates) than the ones demonstrated here would be possible, if the fabricated chip were to include more unit cells. This can be achieved if smaller BULs are employed, which is technically feasible. For practical operation, it is fundamental to thermally stabilize the TBUs so their programmed values remain stable in time once selected and to manage the impact of thermal crosstalk from neighboring TBUs (see Supplementary Note 4). In addition, it is essential to make the chip operation robust against departures of the TBUs from their designed values. Recent works[54, 55] have reported both theoretical and practical solutions to overcome this limitation in CMOS-compatible silicon photonics platforms. Hybrid integration with III–V materials would be necessary in order to incorporate the optical sources and the modulator to the proposed optical core. Future work on electronic integration is required in order to integrate the current sources.

Based on our results, further increasing the number of TBUs to be integrated would require two different metal layer levels to enable on-chip electrical routing. Although a miniaturization of the BUL is possible, we must consider the operational trade-off that relates the maximum attainable free-spectral range, the step resolution and the losses of a synthetized waveguide (see Supplementary Note 7).

Another important issue is related to how the input signal is directed into a specific input port of the mesh network and how a signal is directed from a port of the mesh to a photodetector. These operations require some degree of optical interconnection that can be provided by the mesh itself provided its size is large enough.

In contrast to application-specific devices, multipurpose photonic processors enable a wide variety of applications on the same chip, providing flexible and fast adaptive design topologies and circuit parameters. Fabricated in a CMOS compatible technology, multi-task processors enable high-production volume reducing the price per chip. The unused/available TBUs perform as spare components that will potentially enable self-healing photonic integrated circuits.

In summary, we have designed, fabricated and demonstrated an integrated reconfigurable photonic signal processor. The chip is based on a tuneable hexagonal silicon waveguide mesh configuration where the hexagon sides are composed of two

waveguides, which can be variably coupled or switched by means of a programmable TBU implemented by means of a Mach-Zehnder interferometer. We reconfigured a 7-hexagon cell design to experimentally demonstrate 21 different functionalities, including unbalanced FIR Mach-Zehnder filters, ring cavities, complex CROW, SCISSOR, and ring-loaded MZI filters, and multiple input multiple output linear optic transformation devices, including a C-NOT gate. These devices have applications in a wide variety of fields including communications, biophotonics, sensing, multiprocessor interconnections, switching, and quantum information. Hence, this work represents an important step towards the realization of the new paradigm of multipurpose reconfigurable photonic processors.

## Methods

**Device manufacturing process.** The device was fabricated at the Southampton Nanofabrication Centre at the University of Southampton. SOI wafers with a 220-nm thick silicon overlayer and a 3-μm thick buried oxide layer were used and e-beam lithography performed to define the grating couplers. Dry etching of 70 nm into the silicon overlayer to form the grating couplers was then carried out followed by resist stripping. Another e-beam lithography and 120-nm silicon dry etching step was performed to produce the optical waveguides. Following resist stripping, 1 μm of PECVD silicon dioxide was deposited to act as the upper cladding layer of the waveguides. Photolithography was then performed to define isolation trench openings, followed by a deep dry etching process to etch through the top cladding, silicon overlayer, and buried oxide layer. These trenches provided thermal isolation to adjacent devices and improved the efficiency of the heaters. A 1.8-μm thick metal layer was deposited after the resist had been stripped. A subsequent photolithography and dry etching step realized electrodes used to provide localized heating to tune the devices. The resist was then stripped and the wafers diced into individual dies. These dies were then mounted onto PCBs and a wire bonding process was used to provide electrical connections both within the die and between the die and the PCB

**Device testing process.** We performed a static characterization of the test cell in four different dies to extract information regarding the main optical properties of the integrated waveguides. A tuneable laser (ANDO AQ4321D) featuring a 1 pm wavelength resolution was connected to the input grating coupler of the test cell and scanned to provide a wavelength range characterization. The test cell output grating coupler was connected to an optical spectrum analyzer (ANDO AQ6217C). Measurements included: differential path length to characterize propagation losses, cascaded bends structures to characterize bend losses and two different cascaded and coupled MMI structures to characterize MMI insertion losses and bandwidth. For electro-optical characterization of the TBUs in the 7-cell waveguide mesh, we employed current sources of different resolution and quality: 3 Keihtley2401, 13 Thorlabs LDC8010 and 2 TECMA 72-2535. For the outer perimeter TBUs, the process consisted in injecting optical power into one of the ports of the TBU and sweeping the electrical current bias applied to one of the two heaters. This process was carried out for the 76 thermal tuners present on each of the two characterized PCBs. Altogether with resistance and output optical power, we obtained as a result the normalized coupling constant calibration curves of each TBU. Through this method, we also extracted the phase shift induced by each heater (See Supplementary Note 2). For inner TBU characterization, we biased outer TBUs to set them in a proper state to access the inner devices.

**Data availability.** All data are available from the corresponding author upon reasonable request.

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

## Acknowledgements

J.C. acknowledges funding from the ERC Advanced Grant ERC-ADG-2016-741415 UMWP-Chip, I.G. acknowledges the funding through the Spanish MINECO Ramon y Cajal program. D.P. acknowledges financial support from the UPV through the FPI predoctoral funding scheme. D.J.T. acknowledges funding from the Royal Society for his University Research Fellowship.

## Author contributions

J.C., I.G., and D.P. conceived the processor design. D.P., D.J.T., and G.Z.M. designed the chip. L.C. fabricated the chip, A.Z.K. carried the beam lithography, K.L. the PCB assembly and wirebonding, and W.C. designed the PCB layout. D.P., J.C., and I.G. conceived the experiments and performed the measurements. J.C., I.G., and D.P. analyzed the data and wrote the paper. G.Z.M., D.J.T., I.G., and J.C. managed, coordinated, and supervised the project.

## Additional information

**Competing interests:** The authors declare no competing financial interests.

**Change history:** A correction to this article has been published and is linked from the HTML version of this paper.

