## [Peer Review File · Nature Communications]

Reviewers' comments:

Reviewer #1 (Remarks to the Author):

The revised manuscript includes discussions on algorithms, and provides partial explanations to the comments I have made on the original manuscript, but does not seem to fully address them.

1. Regarding my original questions 1 and 2, the authors provide answers in a) which included the following:

a) " FIR filters are based either on cascades/lattices of 3-dB tunable MZIs or transversal filter configurations. For both alternatives, synthesis and recursive scaling algorithms have been developed in the literature and are available (Refs [42], [43] in the paper). No specific algorithms are required for the hexagonal waveguide mesh if we can show that they can implement either a 3-dB tunable MZI cascade"

The architectures employed in Refs [42] and [43] are different from the architecture in the manuscript. Hence, I do not agree that a possibility of 3 dB tunable MZI cascade is sufficient for proving the existence of synthesis and recursive scaling algorithms. Refs [42] and [43] (and subsequent papers in the literature) also provided details of how poles and zeros can be reconfigured, and how recursive formulas are utilized in multi-stage units. This was important in scalability discussions. On the other hand, the revised manuscript is still incomplete from this perspective. As commented on the original manuscript, "One should note that there are many examples of reconfigurable optical filters that do not have associated synthesis algorithms, and therefore they cannot be used for universal applications." My impression is that this hexagonal architecture may not have well-established synthesis or recursive algorithms.

2. Regarding my original questions 1 and 2, the authors provide answers in b) which included the following:

b) "In this case, detailed synthesis and recursive scaling algorithms have been reported for triangular (References [51] and [36] in the paper) by Reck et al. and Miller, respectively, and rectangular configuration (reference [38] in the paper) by Clements et al. In the supplementary material of the revised version, we demonstrate that both the triangular as well as the rectangular mode transformer configurations can be implemented with the hexagonal waveguide mesh and provide the exact adaptation relationships between the parameters of the TBUs in the hexagonal waveguide mesh design and the MZI devices in both the Reck-Miller and Clements et al. structures. These adaptations were not present in the former version."

Again, the architectures employed in Refs [51] and [36] are different from the architecture in the manuscript, but it appears that the authors are claiming that the architecture in the manuscript can include the architectures in Refs [51] and [36]. Since the author's architecture includes far more (redundant) tunable elements than the architectures in Refs [51] and [36], there will not be a unique solution for each intended unitary operation. This will complicate formulations of synthesis and recursive algorithms.

In summary, this revised manuscript can experimentally show some of the functionalities of previously published universal photonic processors or lattice filters, but has not fully formulated scalable synthesis and recursive algorithms. At the present form, I do not recommend publication of this revised manuscript in Nature Communications.

Reviewer #2 (Remarks to the Author):

The authors have addressed my concerns in the previous review. Some minor suggestions:

1. The title may be revised by adding "signal" before processor, since the processor is a signal processor.
2. In the summary, "an integrated reconfigurable photonic universal processor" may be revised to be "an integrated reconfigurable photonic signal processor core"

I would like to recommend publication of the manuscript in Nature Communications after the minor revisions.

Reviewer #4 (Remarks to the Author):

The new paragraph in the introduction is inconsistent with that in the "Reply to Reviewers". Actually, the one in Reply to Reviewers makes a lot more sense, as it is better written. I suggest that that one is used:

"Zhuang and co-workers³⁹ have pioneered the field by proposing a programmable optical chip architecture connecting MZI devices in a square-shaped mesh network grid. The distinctive feature of this approach is that it enables both feedforward and feedback configurations, selecting the adequate path through the mesh and providing independent tuning of circuit parameters to complex valued coefficients by introducing phase tuning elements in both arms of the MZIs to enable independent control of amplitude and phase of light at coupler outputs^{39,40}".

Point to Point Replies to Referees and List of Changes

Reviewer #1 (Remarks to the Author):

The revised manuscript includes discussions on algorithms, and provides partial explanations to the comments I have made on the original manuscript, but does not seem to fully address them.

1. Regarding my original questions 1 and 2, the authors provide answers in a) which included the following:

a) "FIR filters are based either on cascades/lattices of 3-dB tunable MZIs or transversal filter configurations. For both alternatives, synthesis and recursive scaling algorithms have been developed in the literature and are available (Refs [42], [43] in the paper). No specific algorithms are required for the hexagonal waveguide mesh if we can show that they can implement either a 3-dB tunable MZI cascade"

The architectures employed in Refs [42] and [43] are different from the architecture in the manuscript. Hence, I do not agree that a possibility of 3 dB tunable MZI cascade is sufficient for proving the existence of synthesis and recursive scaling algorithms. Refs [42] and [43] (and subsequent papers in the literature) also provided details of how poles and zeros can be reconfigured, and how recursive formulas are utilized in multi-stage units. This was important in scalability discussions. On the other hand, the revised manuscript is still incomplete from this perspective. As commented on the original manuscript, "One should note that there are many examples of reconfigurable optical filters that do not have associated synthesis algorithms, and therefore they cannot be used for universal applications." My impression is that this hexagonal architecture may not have well-established synthesis or recursive algorithms.

2. Regarding my original questions 1 and 2, the authors provide answers in b) which included the following:

b) "In this case, detailed synthesis and recursive scaling algorithms have been reported for triangular (References [51] and [36] in the paper) by Reck et al. and Miller, respectively, and rectangular configuration (reference [38] in the paper) by Clements et al. In the supplementary material of the revised version, we demonstrate that both the triangular as well as the rectangular mode transformer configurations can be implemented with the hexagonal waveguide mesh and provide the exact adaptation relationships between the parameters of the TBUs in the hexagonal waveguide mesh design and the MZI devices in both the Reck-Miller and Clements et al. structures. These adaptations were not present in the former version."

Again, the architectures employed in Refs [51] and [36] are different from the architecture in the manuscript, but it appears that the authors are claiming that the architecture in the manuscript can include the architectures in Refs [51] and [36]. Since the author's architecture includes far more (redundant) tunable elements than the architectures in Refs [51] and [36], there will not be a unique solution for each intended unitary operation. This will complicate formulations of synthesis and recursive algorithms.

In summary, this revised manuscript can experimentally show some of the functionalities of previously published universal photonic processors or lattice filters, but has not fully formulated scalable synthesis and recursive algorithms. At the present form, I do not recommend publication of this revised manuscript in Nature Communications.

Authors' reply: We thank the reviewer for his comments and share his concerns regarding the issue of the availability of a scalable synthesis algorithm. In this respect, we have been trying to understand his point and in our opinion we believe that there is a misinterpretation of what we are actually reporting, most probably because we were not able to transmit it adequately.

We think that the reviewer is expecting a specific synthesis procedure for the hexagonal waveguide mesh architecture. Our impression is reinforced by some of the sentences in his/her report such as: "My impression is that this hexagonal architecture may not have well-established synthesis or recursive algorithms" or "the architectures employed in Refs [51] and [36] are different from the architecture in the manuscript".

This is actually NOT what we are reporting in the paper. What we are actually reporting is a hardware structure, which can be programmed to **emulate** the specific configurations of both common FIR+IRR filters as well as the two published layouts for triangular and rectangular unitary transforming circuits. We are not claiming that this hardware structure can implement the former circuits in a different way based on a generic synthesis algorithms ad-hoc designed for it. The hexagonal waveguide mesh per se is just a programmable hardware platform supporting multiple configurations and not a specific layout. We are aware that this is a very subtle point and this is the reason why we insist that it works in a similar way as an FPGA in electronics, which not being a specific architecture is programmed to emulate different electronic subsystems.

This leads, therefore, to the question of synthesis algorithms and scalability. Since the mesh emulates particular architectures, then the main point is to show that their synthesis algorithms can be directly translated into specific parameter values of the Mach-Zehnder Interferometers (MZI) that are needed to implement the waveguide coupling points required to emulate a particular structure. This justifies, for example the remark “*there will not be a unique solution for each intended unitary operation*” made by the reviewer as a given unitary operation can be either be implemented by an emulated triangular or an emulated rectangular multiple input/multiple output configuration. Simply stated, yes there can be several solutions to implement an intended unitary operation, each one corresponding to a different structure or layout (i.e triangular or rectangular MZI arrangement), but once the structure that emulates the transformation is chosen then that solution is unique.

Now, the translation equations are direct for typical discrete FIR and IIR filters, while they are more elaborate for universal linear transformers, which have been derived and exposed in detail (including the scalability) in the Supplementary material.

Action performed: We have tried to explain thoroughly this point in the initial part of subsection *Synthesis algorithms and Scalability* inside the *Results* section. The main new material is at the beginning and reads as follows:

The proposed hexagonal waveguide mesh is per se a programmable hardware platform supporting multiple configurations and not a specific layout. This means that it can be programmed to emulate the specific hardware configurations of a wide variety of signal processor architectures, including, among others, traditional feedforward/feedbackward FIR and IIR impulse response filters as well as universal multiple-input/multiple-output optical linear transformers. It is well known from signal processing theory⁴² that general synthesis algorithm that can be applied to obtain, given a particular hardware structure, any kind of transfer function is not available. Therefore not such synthesis algorithm can be developed for the hexagonal waveguide mesh either. However, the available synthesis methods for the specific hardware configurations that can be emulated using the waveguide mesh can be applied by developing a suitable procedure, which translates the results provided by the synthesis equations into specific parameter values of the Mach-Zehnder Interferometers (MZI) that are needed to implement the waveguide coupling points in the emulated layout. We have found that this is possible for all main discrete-time signal processing hardware configurations employed in practice as discussed below, all of which are scalable.

We believe that the former provides a sufficient discussion of the fact that scalable synthesis algorithms exist for the structures that can be emulated by the hexagonal waveguide mesh.

Reviewer #2 (Remarks to the Author):

The authors have addressed my concerns in the previous review. Some minor suggestions:

1. The title may be revised by adding "signal" before processor, since the processor is a signal processor.
2. In the summary, "an integrated reconfigurable photonic universal processor" may be revised to be "an integrated reconfigurable photonic signal processor core"

I would like to recommend publication of the manuscript in Nature Communications after the minor revisions.

Authors' reply: We agree with the suggestion of the reviewer regarding the addition of the word "signal" before processor in the title and the substitution of the word "universal" by the word "signal" in the summary.

Action performed: We have changed the title according to the suggestion from “**General-purpose silicon photonics processor core**” to “**General-purpose silicon photonics signal processor core**”. We have also revised the sentence “**an integrated reconfigurable photonic universal processor**” to “**an integrated reconfigurable photonic signal processor core**” in the summary.

Reviewer #4 (Remarks to the Author):

The new paragraph in the introduction is inconsistent with that in the "Reply to Reviewers". Actually, the one in Reply to Reviewers make a lot more sense, as it is better written. I suggest that that one is used:

“Zhuang and co-workers 39 have pioneered the field by proposing a programmable optical chip architecture connecting MZI devices in a square-shaped mesh network grid. The distinctive feature of this approach is that it enables both feedforward and feedbackward configurations, selecting the adequate path through the mesh and providing independent tuning of circuit parameters to complex valued coefficients by introducing phase tuning elements in both arms of the MZIs to enable independent control of amplitude and phase of light at coupler outputs 39,40”.

Authors' Reply: We agree with the reviewer comment.

Action performed: We have replaced the paragraph in the introduction section as requested.

REVIEWERS' COMMENTS:

Reviewer #1 (Remarks to the Author):

The authors' response to my comments seems to indicate that "they can emulate the specific configurations of both common FIR+IRR filters", and that "this means that it can be programmed to emulate the specific hardware configurations of a wide variety of signal processor architectures." The supplemental section also describes examples. However, the manuscript still fails to provide a complete synthesis algorithm for general-purpose filters. The manuscript falls short of supporting its claim for demonstrating a 'General Purpose Signal Processor', the existence of a synthesis algorithm for such a general purpose signal processor, and the scalability of the synthesis algorithm for such a general purpose signal processor. The fact that emulating and synthesizing some signal processors does not mean that completeness exists for general purpose signal processor synthesis is possible with a well-defined algorithm. The authors should tone down or delete the claims for 'general purpose' and 'scalability' in the manuscript, if those claims cannot be substantiated.

Reply to Reviewer #1

Reviewer #1 (Remarks to the Author):

The authors' response to my comments seems to indicate that "they can emulate the specific configurations of both common FIR+IRR filters", and that "this means that it can be programmed to emulate the specific hardware configurations of a wide variety of signal processor architectures." The supplemental section also describes examples. However, the manuscript still fails to provide a complete synthesis algorithm for general-purpose filters. The manuscript falls short of supporting its claim for demonstrating a 'General Purpose Signal Processor', the existence of a synthesis algorithm for such a general purpose signal processor, and the scalability of the synthesis algorithm for such a general purpose signal processor. The fact that emulating and synthesizing some signal processors does not mean that completeness exists for general purpose signal processor synthesis is possible with a well-defined algorithm. The authors should tone down or delete the claims for 'general purpose' and 'scalability' in the manuscript, if those claims cannot be substantiated.

Authors' reply: We have deleted the claims for '**general purpose**' and '**universality**' in both the title and the text of the main manuscript and the supplementary material. We believe that the new nomenclature "**multipurpose**" is a more precise term as it covers the fact that the structure can emulate and implement a wide variety of signal processing architectures and, at the same time, does not claim a universal or general nature, which, according to the reviewer, can only be justified if a general synthesis algorithm is available for the implementation of any kind of filter.